# Heterogeneous Wasserstein Discrepancy for Incomparable Distributions

## Abstract

Optimal Transport (OT) metrics allow for defining discrepancies between two probability measures. Wasserstein distance is for longer the celebrated OT-distance frequently-used in the literature, which seeks probability distributions to be supported on the *same* metric space. Because of its high computational complexity, several approximate Wasserstein distances have been proposed based on entropy regularization or on slicing, and one-dimensional Wassserstein computation. In this paper, we propose a novel extension of Wasserstein distance to compare two incomparable distributions, that hinges on the idea of *distributional slicing*, embeddings, and on computing the closed-form Wassertein distance between the sliced distributions. We provide a theoretical analysis of this new divergence, called *heterogeneous Wasserstein discrepancy (HWD)*, and we show that it preserves several interesting properties including rotation-invariance. We show that the embeddings involved in HWD can be efficiently learned. Finally, we provide a large set of experiments illustrating the behavior of HWD as a divergence in the context of generative modeling and in query framework.

## 1 Introduction

Optimal Transport-based data analysis has recently found widespread interest in machine learning community, since its significant usefulness to achieve many tasks arising from designing loss functions in supervised learning (Frogner et al., 2015), unsupervised learning (Arjovsky et al., 2017), text classification (Kusner et al., 2015), domain adaptation (Courty et al., 2017), generative models (Arjovsky et al., 2017; Salimans et al., 2018), computer vision (Bonneel et al., 2011; Solomon et al., 2015) among many more applications (Kolouri et al., 2017; Peyré & Cuturi, 2019). Optimal Transport (OT) attempts to match real-world entities through computing distances between distributions, and for that it exploits prior geometric knowledge on the base spaces in which the distributions are valued. Computing OT distance equals to finding the most cost-efficiency way to transport mass from source distribution to target distribution, and it is often referred to as the Monge-Kantorovich or Wasserstein distance (Monge, 1781; Kantorovich, 1942; Villani, 2009).

Matching distributions using Wasserstein distance relies on the assumption that their base spaces must be the same, or that at least a meaningful pairwise distance between the supports of these distributions can be computed. A variant of Wasserstein distance dealing with heterogeneous distributions and overcoming the lack of intrinsic correspondence between their base spaces is Gromov-Wasserstein (GW) distance (Sturm, 2006; Mémoli, 2011). GW distance allows to learn an optimal transport-like plan by measuring how the similarity distances between pairs of supports within each ground space are closed. It is increasingly finding applications for learning problems in shape matching (Mémoli, 2011), graph partitioning and matching (Xu et al., 2019), matching of vocabulary sets between different languages (Alvarez-Melis & Jaakkola, 2018), generative models (Bunne et al., 2019), or matching weighted networks (Chowdhury & Mémoli, 2018). Due to the heterogeneity of the distributions, GW distance uses only the relational aspects in each domain, such as the pairwise relationships to compare the two distributions. As a consequence, the main disadvantage of GW distance is its computational cost as the associated optimization problem is a non-convex quadratic program (Peyré & Cuturi, 2019), and as few as thousand samples can be computationally challenging. Based on the approach of regularized OT (Cuturi, 2013), in which an entropic penalty is added to the original objective function defining the Wasserstein OT problem, Peyré et al. (2016) propose an entropic version called entropic GW discrepancy, that leads to approximate GW distance. Another

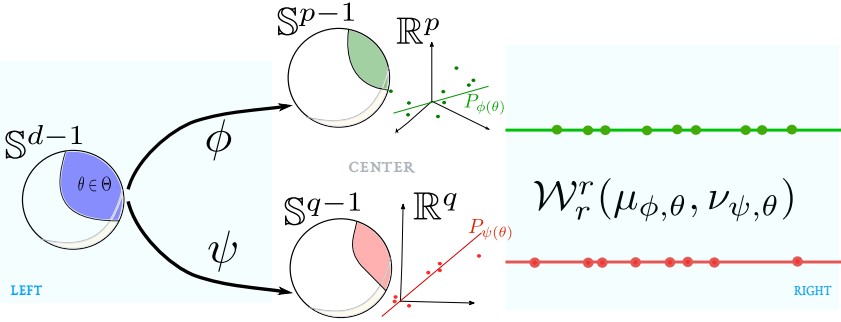

Figure 1: We measure the discrepancy between two distributions living respectively in $\mathbb{R}^p$ and $\mathbb{R}^q$. Our approach is based on generating random slicing projections distributions in each of the metric spaces $\mathbb{R}^p$ and $\mathbb{R}^q$ through the mappings $\phi$ and $\psi$ of a random projection vector sampled from an optimal distribution $\Theta$ in $\mathbb{R}^d$. As each of the projected distribution results in a 1D distribution, we can then compute 1D-Wasserstein distance. It enables us to learn the best projection mappings $\phi$ and $\psi$ and to optimize over the distributional part of the generating projection distribution $\Theta$.

approach for scaling up the GW distance is Sliced Gromov-Wasserstein (SGW) discrepancy (Vayer et al., 2019), which leverages on random projections on 1D and on a closed-form solution of the 1D-Gromov-Wasserstein.

In this paper, we take a different approach for measuring the discrepancy between two heterogeneous distributions. Unlike GW distance that compares pairwise distances of elements from each distribution, we consider a method that embeds the metric measure spaces into a one-dimensional space and computes a Wasserstein distance between the two 1D-projected distributions. The key element of our approach is to learn two mappings that transform vectors from the unit-sphere of a latent space to the unit-sphere of the metric space underlying the two distributions of interest, see Figure 1. In a nutshell, we learn to transform a random direction, sampled under an optimal (learned) distribution (optimality being made clear later), from a $d$-dimensional space to a random direction into the desired spaces. This approach has the benefit of avoiding an ad-hoc padding strategy (completion of 0 of the smaller dimension distributions to fit the high-dimensional one) as in SGW method (Vayer et al., 2019). Another relevant feature of our approach is that the two resulting 1D distributions are now compared through Wasserstein distance. This point, in conjunction, with other key aspect of the method, will lead to a relevant discrepancy between two distributions, called *heterogeneous Wasserstein discrepancy (HWD)*. Although we lose some properties of a distance, we show that HWD is rotation-invariant, that it is robust enough to be considered as a loss for learning generative models between heterogeneous spaces. We also establish that HWD boils down to the recent distributional sliced Wasserstein distance (Nguyen et al., 2020) if the two distributions live in the same space and if some mild constraints are imposed on the mappings.

In summary, our contributions are as follows:

- we propose HWD, a novel slicing-based discrepancy for comparing two distributions living in different spaces. Our chosen formulation is based on comparing 1D random-projected versions of the two distributions using a Wasserstein distance;

- The projection operations are materialized by optimally mapping from one common space to the two spaces of interest. We provide a theoretical analysis of the resulting discrepancy and exhibit its relevant properties;

- Since the discrepancy involves several mappings that need to be optimized, we depict an alternate optimization algorithm for learning them;

- Numerically, we validate the benefits of HWD in terms of comparison between heterogeneous distributions. We show that it can be used as a loss for generative models or shape objects retrieval with better performance and robustness than SGW on those tasks.

## 2    BACKGROUND OF OT DISTANCES

For the reader's convenience, we provide here a brief review of the notations and definitions, that will be frequently used throughout the paper. We start by introducing Wasserstein and Gromov-Wasserstein distances with their sliced versions SW and SGW, where we consider these distances in the specific case of Euclidean base spaces $(\mathbb{R}^p, \|\cdot\|)$ and $(\mathbb{R}^q, \|\cdot\|)$. We denote $\mathscr{P}(\mathcal{X})$ and $\mathscr{P}(\mathcal{Y})$ the respective sets of probability measures whose supports are contained on compact sets $\mathcal{X} \subseteq \mathbb{R}^p$ and $\mathcal{Y} \subseteq \mathbb{R}^q$. For $r \geq 1$, we denote $\mathscr{P}_r(\mathcal{X})$ the subset of measures in $\mathscr{P}(\mathcal{X})$ with finite $r$-th moment ($r \geq 1$), i.e., $\mathscr{P}_r(\mathcal{X}) = \{\eta \in \mathscr{P}(\mathcal{X}) : \int_{\mathcal{X}} \|x\|^r \mathrm{d}\eta(x) < \infty\}$. For $\mu \in \mathscr{P}(\mathcal{X})$ and $\nu \in \mathscr{P}(\mathcal{Y})$, we write $\Pi(\mu, \nu) \subset \mathscr{P}(\mathcal{X} \times \mathcal{Y})$ for the collection of joint probability distributions with marginals $\mu$ and $\nu$, known as couplings, $\Pi(\mu, \nu) = \{\gamma \in \mathscr{P}(\mathcal{X} \times \mathcal{Y}) : \forall A \subset \mathcal{X}, B \subset \mathcal{Y}, \gamma(A \times \mathcal{Y}) = \mu(A), \gamma(\mathcal{X} \times B) = \nu(B)\}$.

### 2.1    OT DISTANCES FOR HOMOGENEOUS DOMAINS

We here assume that the distributions $\mu$ and $\nu$ lie in the same base space, for instance $p = q$. Taking this into account, we can define the Wasserstein distance and its sliced variant.

**Wasserstein distance**    The $r$-th Wasserstein distance is defined on $\mathscr{P}_r(\mathcal{X})$ by

$$\mathcal{W}_r(\mu, \nu) = \left(\inf_{\gamma \in \Pi(\mu, \nu)} \int_{\mathcal{X} \times \mathcal{Y}} \|x - y\|^r \mathrm{d}\gamma(x, y)\right)^{\frac{1}{r}}. \tag{1}$$

The quantity $\mathcal{W}_r(\mu, \nu)$ describes the least amount effort to transform one distribution $\mu$ into another one $\nu$. Since the cost distance used between sample supports is the Euclidean one, the infimum in (1) is attained (Villani, 2009), and any probability $\gamma$ which realizes the minimum is called an *optimal transport plan*. In a finite discrete setting, Problem (1) can be formulated as a linear program, that is challenging to solve algorithmically as its computational cost is of order $\mathcal{O}(n^{5/2} \log n)$ (Lee & Sidford, 2014), where $n$ is the number of sample supports.

Contrastingly, for the 1D case (i.e. $p = 1$) of continuous probability measures, the $r$-th Wasserstein distance has a closed-form solution (Rachev & Rüschendorf, 1998), namely, $\mathcal{W}_r(\mu, \nu) = (\int_0^1 |F_\mu^{-1}(u) - F_\nu^{-1}(u)|^r \mathrm{d}t)^{\frac{1}{r}}$ where $F_\mu^{-1}$ and $F_\nu^{-1}$ are the quantile functions of $\mu$ and $\nu$. For empirical distributions, the 1D-Wasserstein distance is simply calculated by sorting the supports of the distributions on the real line, resulting to a complexity of order $\mathcal{O}(n \log n)$. This nice computational property motivates the use of sliced-Wasserstein (SW) distance (Rabin et al., 2012; Bonneel et al., 2015), where one calculates an (infinity) of 1D-Wasserstein distances between linear projection pushforwards of distributions in question and then computes their average.

To precisely define SW distance, we consider the following notation. Let $\mathbb{S}^{p-1} := \{u \in \mathbb{R}^p : \|u\| = 1\}$ be the unit sphere in $p$ dimension in $\ell_2$-norm, and for any vector $\theta$ in $\mathbb{S}^{p-1}$, we define $P_\theta$ the orthogonal projection onto the real line $\mathbb{R}\theta = \{\alpha\theta : \alpha \in \mathbb{R}\}$, that is $P_\theta(x) = \langle \theta, x \rangle$, where $\langle \cdot, \cdot \rangle$ stands for the Euclidean inner-product. Let $\mu_\theta = P_\theta \# \mu$ the measure on the real line called pushforward of $\mu$ by $P_\theta$, that is $\mu_\theta(A) = \mu(P_\theta^{-1}(A))$ for all Borel set $A \subseteq \mathbb{R}$. We may now define the SW distance.

**Sliced Wasserstein distance**    The $r$-th order sliced Wasserstein distance between two probability distributions $\mu, \nu \in \mathscr{P}_r(\mathcal{X})$ is given by

$$\mathcal{SW}_r(\mu, \nu) = \left(\frac{1}{A_p} \int_{\mathbb{S}^{p-1}} \mathcal{W}_r^r(\mu_\theta, \nu_\theta) \mathrm{d}\theta\right)^{\frac{1}{r}}, \tag{2}$$

where $A_p$ is the area of the surface of $\mathbb{S}^{p-1}$, i.e., $A_p = \frac{2\pi^{p/2}}{\Gamma(p/2)}$ with $\Gamma : \mathbb{R} \to \mathbb{R}$, the Gamma function given as $\Gamma(u) = \int_0^\infty t^{u-1} e^{-t} \mathrm{d}t$. Thanks to its computational benefits and its valid metric property (Bonnotte, 2013), the SW distance has recently been used for OT-based deep generative modeling (Kolouri et al., 2019; Deshpande et al., 2019; Wu et al., 2019). Note that the normalized integral in (2) can be seen as the expectation for $\theta \sim \sigma^{p-1}$, the uniform surface measure on $\mathbb{S}^{p-1}$, that is $\mathcal{SW}_r(\mu, \nu) = (\mathbb{E}_{\theta \sim \sigma^{p-1}}[\mathcal{W}_r^r(\mu_\theta, \nu_\theta)])^{\frac{1}{r}}$. Therefore, the SW distance can be easily approximated via a Monte Carlo sampling scheme by drawing uniform random samples from $\mathbb{S}^{p-1}$:

$\mathcal{SW}_r^r(\mu, \nu) \approx \frac{1}{K} \sum_{k=1}^K \mathcal{W}_r^r(\mu_{\theta_k}, \nu_{\theta_k})$ where $\theta_1, \ldots, \theta_K \overset{i.i.d.}{\sim} \sigma^{p-1}$ and $K$ is the number of random projections.

## 2.2 OT DISTANCES FOR HETEROGENEOUS DOMAINS

To get benefit from the advantages of OT in many machine learning applications involving heterogeneous and incomparable domains ($p \neq q$), the Gromov-Wasserstein distance (Mémoli, 2011) stands for the basic OT distance dealing with this setting.

**Gromov-Wasserstein distance** The $r$-th Gromov-Wasserstein distance between two probability distributions $\mu \in \mathscr{P}_r(\mathcal{X})$ and $\nu \in \mathscr{P}_r(\mathcal{Y})$ is defined by

$$\mathcal{GW}_r(\mu, \nu) = \inf_{\gamma \in \Pi(\mu,\nu)} J_r(\gamma) \overset{\text{def.}}{=} \frac{1}{2} \Big( \iint_{\mathcal{X}^2 \times \mathcal{Y}^2} |\|x - x'\| - \|y - y'\||^r \mathrm{d}\gamma(x, y) \mathrm{d}\gamma(x', y') \Big)^{\frac{1}{r}}. \quad (3)$$

Note that $\mathcal{GW}_r(\mu, \nu)$ is a valid metric endowing the collection of all isomorphism classes metric measure spaces of $\mathscr{P}_r(\mathcal{X}) \times \mathscr{P}_r(\mathcal{Y})$, see Theorem 5 in (Mémoli, 2011). The GW distance learns an optimal transport-like plan which transports samples from a source metric space $\mathcal{X}$ into a target metric space $\mathcal{Y}$, by measuring how the similarity distances between pairs of samples within each space are close. Furthermore, GW distance enjoys several geometric properties, particularly translation and rotation invariance. However, its major bottleneck consists in an expensive computational cost, since problem (3) is non-convex and quadratic. A remedy to such a heavy computational burden lies in an entropic regularized GW discrepancy (Peyré et al., 2016), using Sinkhorn iterations algorithm (Cuturi, 2013). This latter needs a large regularization parameter to guarantee a fast computation, which, unfortunately, entails a poor approximation of the true GW distance value. Another approach to scale up the computation of GW distance is sliced-GW discrepancy (Vayer et al., 2019). The definition of SGW shows 1D-GW distances between projected pushforward of an artifact zero padding of $\mu$ or $\nu$ distribution. We detail this representation in the following paragraph.

**Sliced Gromov-Wasserstein discrepancy** Assume that $p < q$ and let $\Delta$ be an artifact zero padding from $\mathcal{X}$ onto $\mathcal{Y}$, i.e. $\Delta(x) = (x_1, \ldots, x_p, 0, \ldots, 0) \in \mathbb{R}^q$. The $r$-th order sliced Gromov-Wasserstein discrepancy between two probability distributions $\mu \in \mathscr{P}_r(\mathcal{X})$ and $\nu \in \mathscr{P}_r(\mathcal{Y})$ is given by

$$\mathcal{SGW}_{\Delta,r}(\mu, \nu) = \Big( \mathbb{E}_{\theta \sim \sigma^{q-1}} \big[ \mathcal{GW}_r^r((\Delta \# \mu)_\theta, \nu_\theta) \big] \Big)^{\frac{1}{r}}. \quad (4)$$

It is worthy to note that $\mathcal{SGW}_{\Delta,r}$ is depending on the ad-hoc operator $\Delta$, hence the rotation invariance is lost. Vayer et al. (2019) propose a variant of SGW that does not depend on the choice of $\Delta$, called Rotation Invariant SGW (RI-SGW) for $p = q$, defined as the minimizer of $\mathcal{SGW}_{\Delta,r}$ over the Stiefel manifold, see (Vayer et al., 2019, Equation 6). In this work, we are interested in calculating an OT-based discrepancy between distributions over distinct domains using the slicing technique. Our approach is different from the SGW one in many points, specifically (and most importantly) we use a 1D-Wasserstein distance between the projected pushforward distributions and not a 1D-GW distance. In the next section, we detail the setup of our approach.

## 3 HETEROGENEOUS WASSERSTEIN DISCREPANCY

Despite the computational benefit of sliced-OT variant discrepancies, they have an unavoidable bottleneck corresponding to an intractable computation of the expectation with respect to uniform distribution of projections. Furthermore, the Monte Carlo sampling scheme can often generate an overwhelming number of irrelevant directions; hence, the larger number of sample projections, the more accurate approximation of sliced-OT values. Recently, Nguyen et al. (2020) have proposed the *distributional*-SW distance allowing to find an optimal distribution over an expansion area of informative directions. This performs the projection efficiently by choosing an optimal number of important random projections needed to capture the structure of distributions. Our approach for comparing distributions in heterogeneous domains follows a distributional slicing technique combined with OT metric measure embedding (Alaya et al., 2020).

Let us first introduce additional notations. Fix $d \geq 1$ and consider two *nonlinear* mappings $\phi : \mathbb{S}^{d-1} \to \mathbb{S}^{p-1}$ and $\psi : \mathbb{S}^{d-1} \to \mathbb{S}^{q-1}$. For any constants $C_\phi, C_\psi > 0$, we define the following probability measure sets: $\mathscr{M}_{C_\phi} = \left\{ \Theta \in \mathscr{P}(\mathbb{S}^{d-1}) : \mathbb{E}_{\theta, \theta' \sim \Theta}[|\langle \phi(\theta), \phi(\theta') \rangle|] \leq C_\phi \right\}$ and $\mathscr{M}_{C_\psi} = \left\{ \Theta \in \mathscr{P}(\mathbb{S}^{d-1}) : \mathbb{E}_{\theta, \theta' \sim \Theta}[|\langle \psi(\theta), \psi(\theta') \rangle|] \leq C_\psi \right\}$. We say that $C_\phi, C_\psi$ are $(\phi, \psi)$-*admissible* constants if the intersection sets $\mathscr{M}_{C_\phi} \cap \mathscr{M}_{C_\psi}$ is not empty. We hereafter denote $\mu_{\phi,\theta} = P_{\phi(\theta)} \# \mu$ and $\nu_{\psi,\theta} = P_{\psi(\theta)} \# \nu$ the pushforwards of $\mu$ and $\nu$ by projections over unit sphere $P_{\phi(\theta)}$ and $P_{\psi(\theta)}$, respectively.

**Informal presentation** While the distributions $\mu$ and $\nu$ are valued in different spaces, $\mathcal{X} \subset \mathbb{R}^p$ and $\mathcal{Y} \subset \mathbb{R}^q$, any projected distributions will live in real line, enabling the computation of 1D-Wasserstein distance (Figure 1, right). In order to generate random 1D projections in each of the spaces, we map a common random projection distribution from $\mathbb{S}^{d-1}$ into each of the projection spaces $\mathbb{S}^{p-1}$ and $\mathbb{S}^{q-1}$, through the mappings $\phi$ and $\psi$ (see Figure 1, left). Hence, the main components of the *heterogeneous Wasserstein discrepancy* will be the distribution $\Theta \in \mathscr{P}(\mathbb{S}^{d-1})$, and the two embeddings $\phi$ and $\psi$ which will be wisely chosen. The resulting directions $\phi(\theta)$ and $\psi(\theta)$ form the projections $P_{\phi(\theta)}$ and $P_{\psi(\theta)}$ (see Figure 1, center) used to compute several 1D-Wasserstein distances.

## 3.1 DEFINITION AND PROPERTIES

Herein we state the formulation of the proposed discrepancy and exhibit its main theoretical properties.

**Definition 1** *The heterogeneous Wasserstein discrepancy* ($\mathcal{HWD}$) *of order* $r \geq 1$ *between* $\mu \in \mathscr{P}_r(\mathcal{X})$ *and* $\nu \in \mathscr{P}_r(\mathcal{Y})$ *reads as*

$$\mathcal{HWD}_r(\mu, \nu) = \inf_{\phi, \psi} \sup_{\Theta \in \mathscr{M}_{C_\phi} \cap \mathscr{M}_{C_\psi}} \left( \mathbb{E}_{\theta \sim \Theta} \left[ \mathcal{W}_r^r(\mu_{\phi,\theta}, \nu_{\psi,\theta}) \right] \right)^{\frac{1}{r}}. \tag{5}$$

HWD belongs to a family of projected OT works (Paty & Cuturi, 2019; Rowland et al., 2019; Lin et al., 2021) with a particularity for seeking nonlinear projections minimizing a sliced-OT variant. HWD further inherits the distributional slicing benefit by finding an optimal probability measure $\Theta$ of slices on the unit sphere $\mathbb{S}^{d-1}$ coupled with an optimum couple $(\phi, \psi)$ of embeddings. Note that this optimal $\Theta$ verifies the double conditions $\mathbb{E}_{\theta, \theta' \sim \Theta}[|\cos(\phi(\theta), \phi(\theta'))|] \leq C_\phi$ and $\mathbb{E}_{\theta, \theta' \sim \Theta}[|\cos(\psi(\theta), \psi(\theta'))|] \leq C_\psi$. This gives that $C_\phi, C_\psi \leq 1$, hence the sets $\mathscr{M}_{C_\phi}$ and $\mathscr{M}_{C_\psi}$ belong to $\mathscr{M}_1 = \{ \Theta \in \mathscr{P}(\mathbb{S}^{d-1}) \}$ the set of all probability measures of the unit sphere $\mathbb{S}^{d-1}$. It is worthy to note that for small regularizing $(\phi, \psi)$-admissible constants, the measure $\Theta$ is forced to distribute more weights to directions that are far from each other in terms of their angles (Nguyen et al., 2020).

Now, in order to guarantee the existence of $(\phi, \psi)$-admissible constants, we assume that the couple $(\phi, \psi)$-embeddings are *approximately angle preserving*.

**Assumption 1 (Approximately angle preserving property)** *For any couple* $(\phi, \psi)$-*embeddings , assume that there exists two non-negative constants* $L_\phi$ *and* $L_\psi$, *such that the following holds*

$$|\langle \phi(\theta), \phi(\theta') \rangle| \leq L_\phi |\langle \theta, \theta' \rangle| \text{ and } |\langle \psi(\theta), \psi(\theta') \rangle| \leq L_\psi |\langle \theta, \theta' \rangle|, \text{ for all } \theta, \theta' \in \mathbb{S}^{d-1}.$$

In Proposition 1, we deliver lower bounds of the regularizing $(\phi, \psi)$-admissible constants $C_\phi$ and $C_\psi$, depending on the dimension of the latent space $d$ and on the *levels* $(L_\phi, L_\psi)$ of approximately angle preserving property. These bounds ensure the non-emptiness of the sets $\mathscr{M}_{C_\phi}$ and $\mathscr{M}_{C_\psi}$.

**Proposition 1** *Let Assumption 1 hold and consider regularizing* $(\phi, \psi)$-*admissible constants such that* $C_\phi \geq \frac{L_\phi \Gamma(d/2)}{\sqrt{\pi} \Gamma((d+1)/2)}$ *and* $C_\psi \geq \frac{L_\psi \Gamma(d/2)}{\sqrt{\pi} \Gamma((d+1)/2)}$. *Then the sets* $\mathscr{M}_{C_\phi}$ *and* $\mathscr{M}_{C_\psi}$ *contain the uniform measure* $\sigma^{d-1}$ *and* $\bar{\sigma} = \sum_{k=1}^{d} \frac{1}{d} \delta_{\theta_k}$, *where* $\{\theta_1, \dots, \theta_d\}$ *forms any orthonormal basis in* $\mathbb{R}^d$. *Note that by Gautschi's inequality (Gautschi, 1959) for the Gamma function, we have that* $C_\phi \geq \frac{L_\phi}{d}$ *and* $C_\psi \geq \frac{L_\psi}{d}$.

Proof of Proposition 1 is presented in Appendix A.2. Together the admissible constants, the levels of angle preserving property, and the dimension $d$ of the latent space form the hyperparameters set of

HWD problem. For settings of large $d$, the admissible constants could take smaller values, that force the measure $\Theta$ to focus on far-angle directions. However, for smaller $d$, we may lose the control on the distributional part, the set $\mathscr{M}_C$ tends to $\mathscr{M}_1$ the entire set of probability measure on $\mathbb{S}^{d-1}$, hence it boils down on a standard slicing approach that needs an expensive number of projections to get an accurate approximation. Next, we give a set of interesting theoretical properties characterizing HWD.

**Proposition 2** *For any $r \geq 1$, HWD satisfies the following properties:*

(i) *$\mathcal{HWD}_r(\mu, \nu)$ is finite, that is $\mathcal{HWD}_r(\mu, \nu) \leq 2^{\frac{r-1}{r}}(M_r(\mu) + M_r(\nu))$ where $M_r(\cdot)$ is the $r$-th moment of the given distribution, i.e. $M_r(\mu) = \left( \int_{\mathcal{X}} \|x\|^r \mathrm{d}\mu(x) \right)^{\frac{1}{r}}$.*

(ii) *$\mathcal{HWD}_r(\mu, \nu)$ is non-negative, symmetric and verifies $\mathcal{HWD}_r(\mu, \mu) = 0$.*

(iii) *$\mathcal{HWD}_r(\mu, \nu)$ has a discrepancy equivalence given by*

$$\left(\frac{1}{d}\right)^{\frac{1}{r}} \inf_{\phi,\psi} \max_{\theta \in \mathbb{S}^{d-1}} \mathcal{W}_r(\mu_{\phi,\theta}, \nu_{\psi,\theta}) \leq \mathcal{HWD}_r(\mu, \nu) \leq \inf_{\phi,\psi} \max_{\theta \in \mathbb{S}^{d-1}} \mathcal{W}_r(\mu_{\phi,\theta}, \nu_{\psi,\theta}).$$

(iv) *For $p = q$, HWD is upper bounded by the distributional sliced Wasserstein distance.*

(v) *$\mathcal{HWD}_r$ is rotation invariant, namely, $\mathcal{HWD}_r(R\#\mu, Q\#\nu) = \mathcal{HWD}_r(\mu, \nu)$, for any $R \in \mathcal{O}_p = \{R \in \mathbb{R}^{p \times p} : R^\top R = I_p\}$ and $Q \in \mathcal{O}_q = \{Q \in \mathbb{R}^{q \times q} : Q^\top Q = I_q\}$, the orthogonal group of rotations of order $p$ and $q$, respectively.*

(vi) *Let $T_\alpha$ and $T_\beta$ be the translations from $\mathbb{R}^p$ into $\mathbb{R}^p$ and from $\mathbb{R}^q$ into $\mathbb{R}^q$ with vectors $\alpha$ and $\beta$, respectively. Then $\mathcal{HWD}_r(T_\alpha\#\mu, T_\beta\#\nu) \leq 2^{r-1}\left(\mathcal{HWD}_r(\mu, \nu) + \|\alpha\| + \|\beta\|\right)$.*

Proof of Proposition 2 is given in Appendix A.1. From property $(i)$ HWD is finite provided that the distributions in question have a finite $r$-th moments. Note that the supremum over the probability measure sets $\mathscr{M}_{C_\phi}$ and $\mathscr{M}_{C_\psi}$ guarantees the property $\mathcal{HWD}_r(\mu, \mu) = 0$. For $p = q$, if the infimum over the couple $(\phi, \psi)$-embedding in $(iii)$ is realized in the identity mappings, then HWD verifies a metric equivalence with respect to the max-sliced Wasserstein distance (Deshpande et al., 2019). The property $(v)$ highlights a rotation invariance of HWD, which is well verified by the GW distance.

## 3.2 ALGORITHM

Computing HWD requires a resolution of an optimization problem as given in (5). In what follows, we propose an algorithm for computing an approximation of this discrepancy based on samples $X = \{x_i\}_{i=1}^n$ from $\mu$ and samples $Y = \{y_j\}_{j=1}^m$ from $\nu$. At first, let us note that we have min-max optimization to solve; the minimization occuring over the embeddings $\phi$ and $\psi$ and the maximization over the distributions on the unit-sphere. This maximization problem is challenging due to both the constraints and because we optimize over distributions. Similarly to Nguyen et al. (2020), we approximate the problem by replacing the constraints with regularization terms and by replacing the optimization over distributions by an optimization over a push-forward of the uniform probability measure $\sigma^{d-1}$ by a Borel measurable function $f : \mathbb{S}^{d-1} \to \mathbb{S}^{d-1}$. Hence, assuming that we have drawn from a uniform distribution $K$ directions $\{\theta_k\}_{k=1}^K$, the numerical approximation of HWD is obtained by solving the following problem:

$$\min_{\phi,\psi} \quad \max_f \left\{ L_1 \stackrel{\text{def.}}{=} \left( \frac{1}{K} \sum_{k=1}^K \mathcal{W}_r^r \left( X^\top \phi[f(\theta_k)], Y^\top \psi[f(\theta_k)] \right) \right)^{1/r} \right\}$$

$$+ \min_f \lambda_C \left\{ L_2 \stackrel{\text{def.}}{=} \sum_{k,k'} \phi[f(\theta_k)]^\top \phi[f(\theta_{k'})] + \sum_{k,k'} \psi[f(\theta_k)]^\top \psi[f(\theta_{k'})] \right\}$$

$$+ \min_f \lambda_a \left\{ L_3 \stackrel{\text{def.}}{=} \sum_{k,,k'} \left( \phi[f(\theta_k)]^\top \phi[f(\theta_{k'})] - \theta_k^\top \theta_{k'} \right)^2 + \sum_{k,k'} \left( \psi[f(\theta_k)]^\top \psi[f(\theta_{k'})] - \theta_k^\top \theta_{k'} \right)^2 \right\}$$

where the first term in the optimization is related to the sliced Wasserstein, the second term is related to the regularization term associated to $\mathbb{E}_{\theta,\theta'\sim\Theta}[|\langle\phi(\theta), \phi(\theta')\rangle|]$, and $\mathbb{E}_{\theta,\theta'\sim\Theta}[|\langle\psi(\theta), \psi(\theta')\rangle|]$, and the third term is the angle-preserving regularization term. Note that the $\min$ term with respect to $f$ is due to the fact that we want those regularizers to be small. Two hyperparameters $\lambda_C$ and $\lambda_a$

---

**Algorithm 1** COMPUTING HETEROGENEOUS WASSERSTEIN DISCREPANCY (SEE (5))

---

1: **Input:** Source and target samples: $(X, \mu)$ and $(Y, \nu)$; order $r$; the set of random direction $\{\theta_k\}_{k=1}^K$; $T$ number of global iterations; $N$ number of iterations for each of the alternate scheme;
2: **Output:** HWD
3: **function** COMPUTE LOSSES$(X, Y, \phi[f(\theta_k)], \psi[f(\theta_k)])$
4:     compute the average of 1D-Wasserstein $L_1$ between $X^\top \phi[f(\theta_k)]$ and $Y^\top \psi[f(\theta_k)]$
5:     compute the inner-product penalty $L_2$
6:     compute the angle-preserving penalty $L_3$
7: **end function**
8: **for** $t = 1, \cdots, T$ **do**
9:     fix $\phi$ and $\psi$
10:     **for** $i = 1, \cdots, N$ **do**
11:         compute $\phi[f(\theta_k)]$ and $\psi[f(\theta_k)]$
12:         $L_1, L_2, L_3 \leftarrow$ Compute Losses $(X, Y, \phi[f(\theta_k)], \psi[f(\theta_k)])$
13:         $L = -L_1 + L_2 + L_3$
14:         $f \leftarrow f - \gamma_k \nabla L$
15:     **end for**
16:     fix $f$
17:     **for** $i = 1, \cdots, N$ **do**
18:         compute $\phi[f(\theta_k)]$ and $\psi[f(\theta_k)]$
19:         $L_1, L_2, L_3 \leftarrow$ Compute Losses $(X, Y, \phi[f(\theta_k)], \psi[f(\theta_k)])$
20:         $L = L_1 + L_2 + L_3$
21:         $\phi \leftarrow \phi - \gamma_k \nabla L$
22:         $\psi \leftarrow \psi - \gamma_k \nabla L$
23:     **end for**
24: **end for**
25: HWD $\leftarrow$ compute the average over $\{\theta_k\}_{k=1}^K$ of closed-form 1D Wasserstein between $X^\top \phi[f(\theta_k)]$ and $Y^\top \psi[f(\theta_k)]$
26: **Return:** HWD

---

control the impact of these two regularization terms. In practice, $\phi$, $\psi$ and $f$ are parametrized as deep neural networks and the min-max problem is solved by an alternating optimization scheme : (a) optimizing over $f$ with $\psi$ and $\phi$ fixed then (b) optimizing over $\psi$ and $\phi$ with $f$ fixed. Some details of the algorithms are provided in Algorithm 1.

Regarding computational complexity, if we assume that the mapping $f, \phi, \psi$ are already trained, that we have $K$ projections, and that $\phi[f(\theta_k)]$, $\psi[f(\theta_k)]$ are precomputed, then the computation of HWD (line 25 of Algorithm 1) is in $O(K(n \log n + np + nq))$, where $n$ is the number of samples in $X$ and $Y$. When taking into account the full optimization process, then the complexity depends on the number of times we compute the full objective function we are optimizing. Each evaluation requires the computation of the sum in $L_1$ which is $O(K(n \log n + np + nq))$ and the two regularization terms $L_2$ and $L_3$ require both $O(K^2(p + q + 2d))$. Note that in terms of computational complexity, SGW is $O(Kn \log n)$ whereas HWD is $O(TNKn \log n)$, with $T \times N$ being the global number of objective function evaluations. Hence, complexity is in favor of SGW. However, one should note that in practice, because we optimize over the distribution of the random projections, we usually need less slices than SGW and thus depending on the problem, $TNK$ can be of the same magnitude than the number of slices involved in SGW (similar findings have been highlighted for Sliced Wasserstein distance (Nguyen et al., 2020)).

## 4 NUMERICAL EXPERIMENTS

In this section, we analyze HWD, exhibit its rotation-invariant property, and compare its performance with SGW in a generative model context.

**Translation and Rotation** We have used two simple datasets for showing the behavior of HWD with respect to translation and rotation. For translation, we consider two 2D Gaussian distributions one being fixed, the other with varying mean. For rotation, we use two 2D spirals from Scikit-Learn

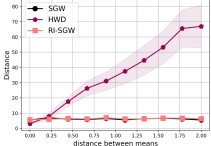 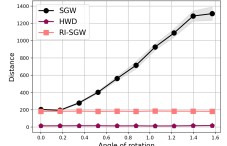 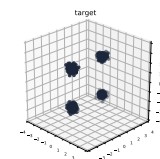 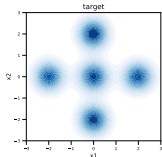

Figure 2: Examples of distance computation between (left) two-translating Gaussian distributions. (right) two spirals.

Figure 3: Examples of target distributions for our generative models (left) 3D 4-mode. (right) 2D 5-mode.

library, one being fixed and the other being rotated from 0 to $\pi/2$. For these two cases, we have drawn 500 samples, used 100 random directions for SGW and RI-SGW. For our HWD, we have used only 10 slices and $T = 50, N = 5$ iterations for each of the alternate optimization. The results we obtain are depicted in Figure 2. From the first panel, we remark that both SGW and RI-SGW are indeed insensitive to translation while HWD captures this translation, which is also verified by property $(vi)$ in Proposition 2. For the spiral problem, as expected HWD and RI-SGW are indeed rotation-invariant while SGW is not.

**Generative models**   For checking whether our distribution discrepancy behaves appropriately, we have used it as a loss function in a generative model. Our task here is to build a model able to generate a distribution defined on a space having a different dimensionality from the target distribution space. As such, we have considered the same toy problems as in Bunne et al. (2019) and investigated two situations: generating 2D distributions from 3D data and the other way around. The 3D target distribution is a Gaussian mixture model with four modes while the 2D ones are 5-mode. Our generative model is composed of a fully-connected neural network with ReLU activation functions. We have considered 3000 samples in the target distributions and batch size of 300. For both HWD and SGW, we have run the algorithm for 30000 iterations with an Adam optimizer, stepsize of 0.001 and default $\beta$ parameters. For the 4-mode problem, the generator is a MLP with 2 layers while for the 5-mode, as the problem is more complex, it has 3 layers. In each case, we have 256 units on the first layer and then 128. For the hyperparameters, we have set $\lambda_C = 1$ and $\lambda_a = 5$ or $\lambda_a = 50$ depending on the problem. Note that for SGW, we have also added a $\ell_2$-norm regularizer on the output of the generator in order to avoid them to drift (see Figures 8 and 9 in Appendix C), as the loss is translation-invariant. Examples of generated distributions are depicted in Figure 4. We remark that our HWD is able to produce visually correct distributions whereas SGW struggles in generating the 4 modes and its 3D 5-mode is squeezed on its third dimension.

**Scalability**   We consider the non-rigid shape world dataset (Bronstein et al., 2006) which consists of 148 three-dimensional shapes from 12 classes. We draw randomly $n \in \{100, 250, 500, 1000, 1500, 2000\}$ vertices $\{x_i \in \mathbb{R}^3\}_{i=1}^n$ on each shape and use them to measure the similarity between a pair of shapes $\{x_i \in \mathbb{R}^3\}_{i=1}^n$ and $\{y_j \in \mathbb{R}^3\}_{j=1}^n$. Figure 5 reports the average time to compute on a single core such a similarity for 100 pairs of shapes using respectively GW, SGW and HWD. As expected GW exhibits a slow behavior while the computational burden of HWD is on par with SGW.

**Classification under various transformations**   This experiment, whose details are provided in Appendix B.2, aims to evaluate the robustness of SGW and HWD (the computationally efficient methods) to different transformations in terms of classification accuracy. To that purpose we employ the Shape Retrieval Contest (SHREC'2010) correspondence dataset, see Bronstein et al. (2010). It includes high resolution (10K-50K) triangular meshes. The shapes are of 3 classes (see Figure 10 in Appendix C) with 9 different transformations and the null shape (no transformation). Each transformation is applied up to five strength levels (weak to strong). Along with the null shape, we consider all strengths of the "isometry", "topology", "scale", "shotnoise" transformations leading to 63 samples. We perform a 1-NN classification. Obtained performances over 10 runs are depicted

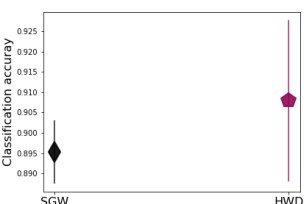

Figure 6: Classification performance under transformations.

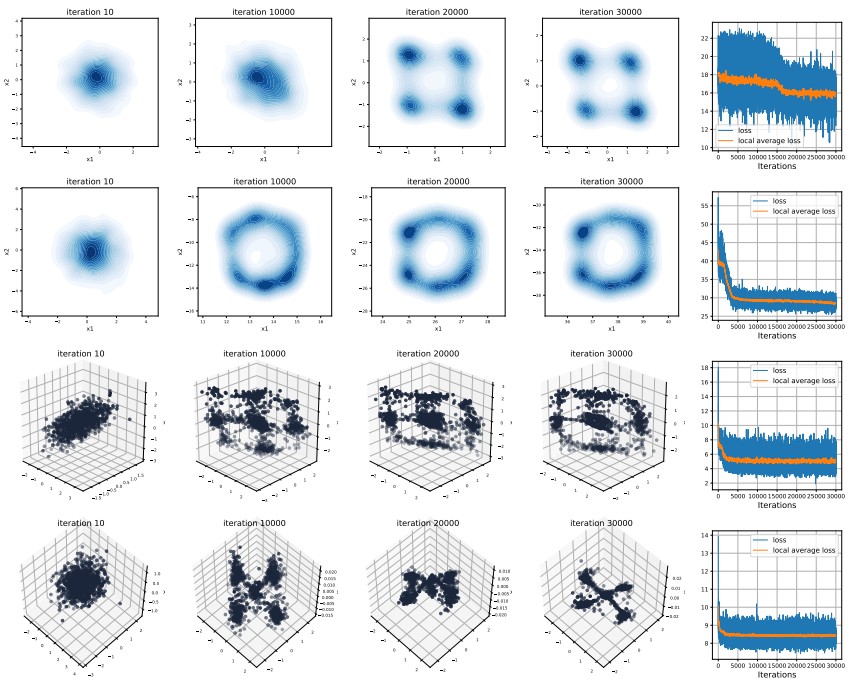

Figure 4: Examples of generated distributions across iterations (10, 10000, 20000, and 30000) for two targets.From top to bottom (first-row) HWD for the 4-mode. (second-row) SGW for the 4-mode (third-row) HWD for 3D 5-mode. (fourth-row) SGW for 5-mode. For each row, the last panel shows the evolution of the loss over the 30000 iterations.

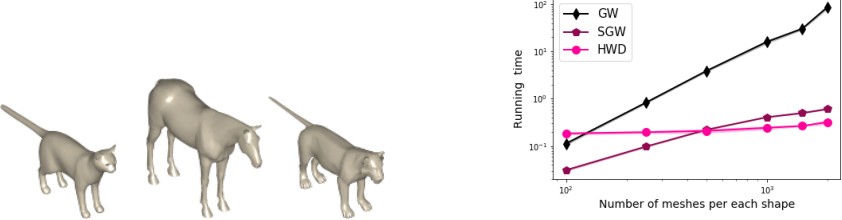

Figure 5: Computation time with respect to $n$, the number of vertices on each shape. (Left-panel) Instances of 3D objects. (Right-panel) Running time.

in Figure 6. They highlight the ability of HWD to be robust to perturbations. HWD achieves slightly better mean classification accuracy than SGW with a competitive computation time (see Figure 5). Notice that GW and RISGW are unable to run under reasonable time-budget constraint.

## 5 CONCLUSION

We introduce in this paper HWD a novel OT-based discrepancy between distributions lying in different spaces. It takes computational benefits from distributional slicing technique, which amounts to find an optimal number of random projections needed to capture the structure of data distributions. Another feature of this discrepancy consists in projecting the distributions in question through a learning of embeddings enjoying the same latent space. We showed a nice geometrical property verified by the proposed discrepancy, specifically a rotation-invariance. We illustrated through extensive experiments the applicability of this discrepancy on generative modeling and shape objects retrieval. We argue that the implementation part faces the standard deep learning bottleneck of tuning the model's hyperparameters. A future extension line of this work is to deliver theoretical guarantees regarding the regularizing parameters, both of distributional and angle preserving properties.

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

# A    PROOFS

## A.1    PROOF OF PROPOSITION 1

We use the following result:

**Lemma 1** *[Theorem 3 in (Nguyen et al., 2020)] For uniform measure $\sigma^{d-1}$ on the unit sphere $\mathbb{S}^{d-1}$, we have*

$$\int_{\mathbb{S}^{d-1}\times\mathbb{S}^{d-1}} |\langle\theta,\theta'\rangle| \mathrm{d}\sigma^{d-1}(\theta)\mathrm{d}\sigma^{d-1}(\theta') = \frac{\Gamma(d/2)}{\sqrt{\pi}\Gamma((d+1)/2)}.$$

Hence,

$$\int_{\mathbb{S}^{d-1}\times\mathbb{S}^{d-1}} |\langle\phi(\theta),\phi(\theta')\rangle| \mathrm{d}\sigma^{d-1}(\theta)\mathrm{d}\sigma^{d-1}(\theta') \overset{\text{(Assumption 1)}}{\leq} L_\phi \int_{\mathbb{S}^{d-1}\times\mathbb{S}^{d-1}} |\langle\theta,\theta'\rangle| \mathrm{d}\sigma^{d-1}(\theta)\mathrm{d}\sigma^{d-1}(\theta')$$

$$\overset{\text{(Lemma 1)}}{\leq} \frac{L_\phi\Gamma(d/2)}{\sqrt{\pi}\Gamma((d+1)/2)}.$$

Therefore, as long as the $(\phi,\psi)$-admissible constants $C_\phi \geq \frac{L_\phi\Gamma(d/2)}{\sqrt{\pi}\Gamma((d+1)/2)}$ and $C_\psi \geq \frac{L_\psi\Gamma(d/2)}{\sqrt{\pi}\Gamma((d+1)/2)}$, we have $\sigma^{d-1} \in \mathscr{M}_{C_\phi} \cap \mathscr{M}_{C_\psi}$. Now using a Gautschi's inequality (Gautschi, 1959) for the Gamma function, it yields that $\frac{\Gamma(d/2)}{\sqrt{\pi}\Gamma((d+1)/2)} \geq \frac{1}{\sqrt{\pi(d+1)/2}} \geq 1/d$. Let $\bar{\sigma} = \sum_{l=1}^{d} \frac{1}{d}\delta_{\theta_l}$, where $\{\theta_1,\ldots,\theta_d\}$ form an orthonormal basis in $\mathbb{R}^d$. We then have

$$\mathbb{E}_{\theta,\theta'\sim\bar{\sigma}}\big[|\langle\phi(\theta),\phi(\theta')\rangle|\big] = \sum_{1\leq k,l\leq d} \Big(\frac{1}{d}\Big)^2 |\langle\phi(\theta_k),\phi(\theta'_l)\rangle| \overset{\text{(Assumption 1)}}{\leq} L_\phi \sum_{1\leq k,l\leq d} \Big(\frac{1}{d}\Big)^2 |\langle\theta_k,\theta'_l\rangle| = \frac{L_\phi}{d}.$$

Therefore we get the lower bounds for the $(\phi,\psi)$-admissible constants $C_\phi$ and $C_\psi$ given in Proposition 1, that guarantee $\sigma^{d-1}, \bar{\sigma} \in \mathscr{M}_{C_\phi} \cap \mathscr{M}_{C_\psi}$.

## A.2    PROOF OF PROPOSITION 2

Let us first state the two following lemmas: Lemma 2 writes an integration result using push-forward measures; it relates integrals with respect to a measure $\eta$ and its push-forward under a measurable map $f : \mathcal{X} \to \mathcal{Y}$. Lemma 3 proves that the admissible set of couplings between the embedded measures are exactly the embedded of the admissible couplings between the original measures.

**Lemma 2** *[See Lerner (2014) p. 61] Let $f : S \to T$ be a measurable mapping, let $\eta$ be a measurable measure on $S$, and let $g$ be a measurable function on $T$. Then $\int_T g \mathrm{d}f_\# \eta = \int_S (g \circ f)\mathrm{d}\eta$.*

**Lemma 3** *[Lemma 6 in Paty & Cuturi (2019)] For all $\phi,\psi$ and $\mu \in \mathscr{P}(\mathcal{X}), \nu \in \mathscr{P}(\mathcal{Y})$, one has $\Pi(\phi\#\mu,\psi\#\nu) = \{(\phi \otimes \psi)\#\gamma \text{ s.t. } \gamma \in \Pi(\mu,\nu)\}$, where $\phi \otimes \psi : \mathcal{X} \times \mathcal{Y} \to \mathcal{X} \times \mathcal{Y}$ such that $(\phi \otimes \psi(x,y) = (\phi(x),\psi(y))$ for all $x,y \in \mathcal{X} \times \mathcal{Y}$.*

● (i) $\mathcal{HWD}_r(\mu,\mu)$ is finite. In one hand, we assume that $\mu \in \mathscr{P}_r(\mathcal{X})$ and $\nu \in \mathscr{P}_r(\mathcal{Y})$, hence its $r$-th moments are finite, i.e., $M_r(\mu) = \big(\int_{\mathcal{X}} \|x\|^r \mathrm{d}\mu(x)\big)^{1/r} < \infty$ and $M_r(\nu) = \big(\int_{\mathcal{Y}} \|y\|^r \mathrm{d}\nu(y)\big)^{1/r} < \infty$. In the other hand, the following holds for all parameter $\theta \in \mathbb{S}^{d-1}$ and a couple $(\phi,\psi)$-embeddings,

$$\mathcal{W}_r^r(\mu_{\phi,\theta},\nu_{\psi,\theta}) = \inf_{\pi\in\Pi(P_{\phi(\theta)}\#\mu,\psi(\theta)\#\nu)} \int_{\mathbb{R}\times\mathbb{R}} |u-u'|^r \mathrm{d}\pi(u,u')$$

$$\overset{\text{(Lemma 3)}}{=} \inf_{\gamma\in\Pi(\mu,\nu)} \int_{\mathcal{X}\times\mathcal{Y}} |\phi(\theta)^\top x - \psi(\theta)^\top y|^r \mathrm{d}\gamma(x,y)$$

$$\leq 2^{r-1} \inf_{\gamma\in\Pi(\mu,\nu)} \int_{\mathcal{X}\times\mathcal{Y}} \big(|\phi(\theta)^\top x|^r + |\psi(\theta)^\top y|^r\big)\mathrm{d}\gamma(x,y)$$

$$= 2^{r-1} \inf_{\gamma\in\Pi(\mu,\nu)} \Big(\int_{\mathcal{X}} |\phi(\theta)^\top x|^r \mathrm{d}\mu(x) + \int_{\mathcal{Y}} |\psi(\theta)^\top y|^r \mathrm{d}\nu(y)\Big),$$

where we use the facts that $(s + t)^r \leq 2^{r-1}(s^r + t^r), \forall s, t \in \mathbb{R}_+$ and that any $\gamma$ transport plan has marginals $\mu$ on $\mathcal{X}$ and $\nu$ on $\mathcal{Y}$. By Cauchy–Schwarz inequality, we get

$$\mathcal{W}_r^r(\mu_{\phi,\theta}, \nu_{\psi,\theta}) \leq 2^{r-1}\Big( \int_{\mathcal{X}} \|\phi(\theta)\|^r \|x\|^r \mathrm{d}\mu(x) + \int_{\mathcal{Y}} \|\psi(\theta)\|^r \|y\|^r \mathrm{d}\nu(y) \Big) = 2^{r-1}\big( M_r^r(\mu) + M_r^r(\nu) \big).$$

Then, $\big( \mathbb{E}_{\theta \sim \Theta} [\mathcal{W}_r^r(\mu_{\phi,\theta}, \nu_{\psi,\theta})] \big)^{1/r} \leq 2^{\frac{r-1}{r}} \big( M_r^r(\mu) + M_r^r(\nu) \big)^{1/r} \leq 2^{\frac{r-1}{r}} \big( M_r(\mu) + M_r(\nu) \big)$. Finally, one has that $\mathcal{HWD}_r(\mu, \nu) \leq 2^{\frac{r-1}{r}} \big( M_r(\mu) + M_r(\nu) \big)$.

● $(ii)$ *Non-negativity and symmetry.* Together the non-negativity, symmetry of Wasserstein distance and the decoupling property of iterated infima (or principle of the iterated infima) yield the non-negativity and symmetry of the distributional sliced sub-embedding distance.

● $(ii)$ $\mathcal{HWD}_r(\mu, \mu) = 0$. Let $\phi$ and $\phi'$ two embeddings for projecting the same distribution $\mu$. Without loss of generality, we suppose that the corresponding $(\phi, \phi')$-admissible constants $C'_\phi \leq C_\phi$, hence $\mathscr{M}_{C_{\phi'}} \subseteq \mathscr{M}_{C_\phi}$. Using the fact that $\sup(A \cap B) \leq \sup A \wedge \sup B$, (with $a \wedge b) = \min(a, b)$), se have, straightforwardly,

$$\mathcal{HWD}_r(\mu, \mu) = \inf_{\phi,\phi'} \sup_{\Theta \in \mathscr{M}_{C_\phi} \cap \mathscr{M}_{C_{\phi'}}} \Big( \mathbb{E}_{\theta \sim \Theta} \big[ \mathcal{W}_r^r(\mu_{\phi,\theta}, \mu_{\phi',\theta}) \big] \Big)^{\frac{1}{r}}$$

$$\leq \inf_{\phi,\phi'} \Big( \sup_{\Theta \in \mathscr{M}_{C_\phi}} \big( \mathbb{E}_{\theta \sim \Theta} \big[ \mathcal{W}_r^r(\mu_{\phi,\theta}, \mu_{\phi',\theta}) \big] \big)^{\frac{1}{r}} \wedge \sup_{\Theta \in \mathscr{M}_{C_{\phi'}}} \big( \mathbb{E}_{\theta \sim \Theta} \big[ \mathcal{W}_r^r(\mu_{\phi,\theta}, \mu_{\phi',\theta}) \big] \big)^{\frac{1}{r}} \Big)$$

$$= \inf_\phi \inf_{\phi'} \Big( \sup_{\Theta \in \mathscr{M}_{C_\phi}} \big( \mathbb{E}_{\theta \sim \Theta} \big[ \mathcal{W}_r^r(\mu_{\phi,\theta}, \mu_{\phi',\theta}) \big] \big)^{\frac{1}{r}} \wedge \sup_{\Theta \in \mathscr{M}_{C_{\phi'}}} \big( \mathbb{E}_{\theta \sim \Theta} \big[ \mathcal{W}_r^r(\mu_{\phi,\theta}, \mu_{\phi',\theta}) \big] \big)^{\frac{1}{r}} \Big)$$

$$\leq \inf_\phi \Big( \sup_{\Theta \in \mathscr{M}_{C_\phi}} \big( \mathbb{E}_{\theta \sim \Theta} \big[ \mathcal{W}_r^r(\mu_{\phi,\theta}, \mu_{\phi,\theta}) \big] \big)^{\frac{1}{r}} \wedge \sup_{\Theta \in \mathscr{M}_{C_{\phi'}}} \big( \mathbb{E}_{\theta \sim \Theta} \big[ \mathcal{W}_r^r(\mu_{\phi,\theta}, \mu_{\phi,\theta}) \big] \big)^{\frac{1}{r}} \Big)$$

$$\leq \inf_\phi \sup_{\Theta \in \mathscr{M}_{C_\phi}} \Big( \mathbb{E}_{\theta \sim \Theta} \big[ \mathcal{W}_r^r(\mu_{\phi,\theta}, \mu_{\phi,\theta}) \big] \Big)^{\frac{1}{r}}$$

$$= 0.$$

● $(iii)$ One has $\big( \frac{1}{d} \big)^{\frac{1}{r}} \inf_{\phi,\psi} \max_{\theta \in \mathbb{S}^{d-1}} \mathcal{W}_r(\mu_{\phi,\theta}, \nu_{\psi,\theta}) \leq \mathcal{HWD}_r(\mu, \nu) \leq \inf_{\phi,\psi} \max_{\theta \in \mathbb{S}^{d-1}} \mathcal{W}_r(\mu_{\phi,\theta}, \nu_{\psi,\theta})$. Since $\mathscr{M}_{C_\phi} \cap \mathscr{M}_{C_\psi} \subset \mathscr{M}_1$ and $\mathcal{W}_r^r(\mu_{\phi,\theta}, \nu_{\psi,\theta}) \leq \max_{\theta \in \mathbb{S}^{d-1}} \mathcal{W}_r^r(\mu_{\phi,\theta}, \nu_{\psi,\theta})$ we find that

$$\sup_{\Theta \in \mathscr{M}_{C_\phi} \cap \mathscr{M}_{C_\psi}} \Big( \mathbb{E}_{\theta \sim \Theta} \big[ \mathcal{W}_r^r(\mu_{\phi,\theta}, \nu_{\psi,\theta}) \big] \Big)^{\frac{1}{r}} \leq \sup_{\Theta \in \mathscr{M}_1} \Big( \mathbb{E}_{\theta \sim \Theta} \big[ \mathcal{W}_r^r(\mu_{\phi,\theta}, \nu_{\psi,\theta}) \big] \Big)^{\frac{1}{r}}$$

$$\leq \big( \max_{\theta \in \mathbb{S}^{d-1}} \mathcal{W}_r^r(\mu_{\phi,\theta}, \nu_{\psi,\theta}) \big)^{1/r}$$

$$\leq \max_{\theta \in \mathbb{S}^{d-1}} \mathcal{W}_r(\mu_{\phi,\theta}, \nu_{\psi,\theta}),$$

which entails that $\mathcal{HWD}_r(\mu, \nu) \leq \inf_{\phi,\psi} \max_{\theta \in \mathbb{S}^{d-1}} \mathcal{W}_r(\mu_{\phi,\theta}, \nu_{\psi,\theta})$. Moreover, since the $(\phi, \psi)$-admissible constants $C_\phi$ and $C_\psi$ satisfy $C_\phi \geq \frac{U_\phi}{d}$ and $C_\psi \geq \frac{U_\psi}{d}$ hence $\bar{\sigma} = \sum_{l=1}^d \frac{1}{d} \delta_{\theta_l} \in \mathscr{M}_{C_\phi} \cap \mathscr{M}_{C_\psi}$, where we set $\theta_1 = \operatorname{argmax}_{\theta \in \mathbb{S}^{d-1}} \mathcal{W}_r(\mu_{\phi,\theta}, \nu_{\psi,\theta})$. We then obtain

$$\mathcal{HWD}_r(\mu, \nu) \geq \inf_{\phi,\psi} \Big( \mathbb{E}_{\theta \sim \bar{\sigma}} \big[ \mathcal{W}_r^r(\mu_{\phi,\theta}, \nu_{\psi,\theta}) \big] \Big)^{\frac{1}{r}}$$

$$= \inf_{\phi,\psi} \Big( \sum_{l=1}^d \frac{1}{d} \mathcal{W}_r^r(\mu_{\phi,\theta_l}, \nu_{\psi,\theta_l}) \Big)^{\frac{1}{r}}$$

$$\geq \big( \frac{1}{d} \big)^{1/r} \inf_{\phi,\psi} \mathcal{W}_r(\mu_{\phi,\theta_1}, \nu_{\psi,\theta_1})$$

$$= \big( \frac{1}{d} \big)^{1/r} \inf_{\phi,\psi} \max_{\theta \in \mathbb{S}^{d-1}} \mathcal{W}_r(\mu_{\phi,\theta}, \nu_{\psi,\theta}).$$

• $(iv)$ *For $p = q$, HWD is upper bound by the distributional Wasserstein distance (DSW)* . Let us first recall the DSW distance: let $C > 0$ and set $\mathscr{M}_C = \{\Theta \in \mathscr{P}(\mathbb{S}^{d-1}) : \mathbb{E}_{\theta,\theta' \sim \Theta}[|\langle \theta, \theta' \rangle|] \leq C\}$.

$$\mathcal{DSW}_r(\mu, \nu) = \sup_{\Theta \in \mathscr{M}_C} \left( \mathbb{E}_{\theta \sim \Theta}\left[ \mathcal{W}_r^r(\mu_\theta, \nu_\theta) \right] \right)^{\frac{1}{r}}.$$

We have that the case of a identity couple of embeddings, $\phi = Id, \psi = Id$, the probability measure set $\mathscr{M}_{C_\phi}, \mathscr{M}_{C_\psi} = \mathscr{M}_C$, then it is trivial that $\mathcal{HWD}_r(\mu, \nu) \leq \mathcal{DSW}_r(\mu, \nu)$.

• $(v)$ *Rotation invariance.* Note that $(R\#\mu)_{\phi,\theta} = P_{\phi(\theta)}\#(R\#\mu) = (P_{\phi(\theta)} \circ R)\#\mu$, and for all $x \in \mathbb{R}^p$, using the adjoint operator $R^*$, $(R^* = R^{-1})$, $(P_{\phi(\theta)} \circ R)(x) = \langle \phi(\theta), R(x) \rangle = \langle R^*(\phi(\theta)), x \rangle = P_{R^* \circ \phi(\theta)}(x)$. Then, $(R\#\mu)_{\phi,\theta} = (P_{R^* \circ \phi(\theta)})\#\mu$. Analogously, one has $(Q\#\nu)_{\psi,\theta} = (P_{Q^* \circ \psi(\theta)})\#\nu$. Moreover,

$$\begin{aligned}
\mathscr{M}_{C_\phi} &= \left\{ \Theta \in \mathscr{P}(\mathbb{S}^{d-1}) : \mathbb{E}_{\theta,\theta' \sim \Theta}[|\langle \phi(\theta), \phi(\theta') \rangle|] \right\} \\
&= \left\{ \Theta \in \mathscr{P}(\mathbb{S}^{d-1}) : \mathbb{E}_{\theta,\theta' \sim \Theta}[|\langle (R^* \circ \phi)(\theta), (R^* \circ \phi)(\theta') \rangle|] \right\} \\
&= \mathscr{M}_{C_{R^* \circ \phi}}.
\end{aligned}$$

Then we have similarly $\mathscr{M}_{C_\psi} = \mathscr{M}_{C_{Q^* \circ \psi}}$. This implies

$$\begin{aligned}
\mathcal{HWD}_r(R\#\mu, Q\#\nu) &= \inf_{\phi,\psi} \sup_{\Theta \in \mathscr{M}_{C_\phi} \cap \mathscr{M}_{C_\psi}} \left( \mathbb{E}_{\theta \sim \Theta}\left[ \mathcal{W}_r^r((R\#\mu)_{\phi,\theta}, (Q\#\nu)_{\psi,\theta}) \right] \right)^{\frac{1}{r}} \\
&= \inf_{\phi,\psi} \sup_{\Theta \in \mathscr{M}_{C_\phi} \cap \mathscr{M}_{C_\psi}} \left( \mathbb{E}_{\theta \sim \Theta}\left[ \mathcal{W}_r^r((P_{R^* \circ \phi(\theta)})\#\mu, P_{Q^* \circ \psi(\theta)}\#\nu) \right] \right)^{\frac{1}{r}} \\
&= \inf_{\phi,\psi} \sup_{\Theta \in \mathscr{M}_{C_\phi} \cap \mathscr{M}_{C_\psi}} \left( \mathbb{E}_{\theta \sim \Theta}\left[ \mathcal{W}_r^r(\mu_{R^* \circ \phi,\theta}, \mu_{R^* \circ \phi,\theta}) \right] \right)^{\frac{1}{r}} \\
&= \inf_{\phi,\psi} \sup_{\Theta \in \mathscr{M}_{C_{R^* \circ \phi}} \cap \mathscr{M}_{C_{Q^* \circ \psi}}} \left( \mathbb{E}_{\theta \sim \Theta}\left[ \mathcal{W}_r^r(\mu_{R^* \circ \phi,\theta}, \nu_{Q^* \circ \phi,\theta}) \right] \right)^{\frac{1}{r}} \\
&= \inf_{\phi,\psi} \sup_{\Theta \in \mathscr{M}_{C_{R^* \circ \phi}} \cap \mathscr{M}_{C_{Q^* \circ \psi}}} \left( \mathbb{E}_{\theta \sim \Theta}\left[ \mathcal{W}_r^r(\mu_{R^* \circ \phi,\theta}, \nu_{Q^* \circ \psi,\theta}) \right] \right)^{\frac{1}{r}} \\
&= \inf_{\phi'=R^* \circ \phi, \psi'=Q^* \circ \psi} \sup_{\Theta \in \mathscr{M}_{C_{\phi'}} \cap \mathscr{M}_{C_{\psi'}}} \left( \mathbb{E}_{\theta \sim \Theta}\left[ \mathcal{W}_r^r(\mu_{\phi',\theta}, \nu_{\psi',\theta}) \right] \right)^{\frac{1}{r}} \\
&= \mathcal{HWD}_r(\mu, \nu).
\end{aligned}$$

• $(vi)$ *Translation quasi-invariance.* We have

$$\mathcal{HWD}_r(T_\alpha\#\mu, T_\beta\#\nu) = \inf_{\phi,\psi} \sup_{\Theta \in \mathscr{M}_{C_\phi} \cap \mathscr{M}_{C_\psi}} \left( \mathbb{E}_{\theta \sim \Theta}\left[ \mathcal{W}_r^r((T_\alpha\#\mu)_{\phi,\theta}, (T_\beta\#\nu)_{\psi,\theta}) \right] \right)^{\frac{1}{r}}.$$

By Lemmas 3 and 2 , we have

$$\begin{aligned}
&\mathcal{W}_r^r((T_\alpha\#\mu)_{\phi,\theta}, (T_\beta\#\nu)_{\psi,\theta}) \\
&\quad = \inf_{\gamma \in \Pi((T_\alpha\#\mu)_{\phi,\theta}, (T_\beta\#\nu)_{\psi,\theta}))} \int_{\mathbb{R}^2} |u - v|^r \mathrm{d}\gamma(u, v) \\
&\quad = \inf_{\gamma \in \Pi((P_{\phi(\theta)} \circ T_\alpha)\#\mu, (P_{\psi(\theta)} \circ T_\beta)\#\nu)} \int_{\mathbb{R}^2} |u - v|^r \mathrm{d}\gamma(u, v) \\
&\quad = \inf_{\gamma \in \Pi(\mu, \nu)} \int_{\mathcal{X} \times \mathcal{Y}} |P_{\phi(\theta)} \circ T_\alpha(x) - P_{\psi(\theta)} \circ T_\beta)(y)|^r \mathrm{d}\gamma(x, y) \\
&\quad = \inf_{\gamma \in \Pi(\mu, \nu)} \int_{\mathcal{X} \times \mathcal{Y}} |(P_{\phi(\theta)}(x) - P_{\psi(\theta)}(y)) + (P_{\phi(\theta)}(\alpha) - P_{\psi(\theta)}(\beta))|^r \mathrm{d}\gamma(x, y) \\
&\quad \leq 2^{r-1}\left( \inf_{\gamma \in \Pi(\mu, \nu)} \int_{\mathcal{X} \times \mathcal{Y}} |(P_{\phi(\theta)}(x) - P_{\psi(\theta)}(y))|^r \mathrm{d}\gamma(x, y) + |P_{\phi(\theta)}(\alpha) - P_{\psi(\theta)}(\beta)|^r \right) \\
&\quad \leq 2^{r-1}\left( \inf_{\gamma \in \Pi(\mu, \nu)} \int_{\mathcal{X} \times \mathcal{Y}} |(P_{\phi(\theta)}(x) - P_{\psi(\theta)}(y))|^r \mathrm{d}\gamma(x, y) + (\|\alpha\| + \|\beta\|)^r \right).
\end{aligned}$$

Thanks to Minkowski inequality,

$$\sup_{\Theta \in \mathscr{M}_{C_\phi} \cap \mathscr{M}_{C_\psi}} \left( \mathbb{E}_{\theta \sim \Theta} \left[ \mathcal{W}_r^r((T_\alpha \# \mu)_\theta^\phi, (T_\beta \# \nu)_\theta^\psi)] \right) \right)^{\frac{1}{r}}$$

$$\leq 2^{r-1} \sup_{\Theta \in \mathscr{M}_{C_\phi} \cap \mathscr{M}_{C_\psi}} \left( \mathbb{E}_{\theta \sim \Theta} \left[ \inf_{\gamma \in \Pi(\mu,\nu)} \int_{\mathcal{X} \times \mathcal{Y}} |(P_{\phi(\theta)}(x) - P_{\psi(\theta)}(y))|^r \mathrm{d}\gamma(x, y) \right] \right)^{\frac{1}{r}}$$

$$+ 2^{r-1} (\|\alpha\| + \|\beta\|) \sup_{\Theta \in \mathscr{M}_{C_\phi} \cap \mathscr{M}_{C_\psi}} \left( \Theta(\mathbb{S}^{d-1}) \right)^{\frac{1}{r}}$$

$$\leq 2^{r-1} \sup_{\Theta \in \mathscr{M}_{C_\phi} \cap \mathscr{M}_{C_\psi}} \left( \mathbb{E}_{\theta \sim \Theta} \left[ \mathcal{W}_r^r(\mu_{\phi,\theta}, \nu_{\psi,\theta})] \right) \right)^{\frac{1}{r}} + 2^{r-1} (\|\alpha\| + \|\beta\|).$$

Therefore, we get $\mathcal{HWD}_r(T_\alpha \# \mu, T_\beta \# \nu) \leq 2^{r-1} \mathcal{HWD}_r(\mu, \nu) + 2^{r-1}(\|\alpha\| + \|\beta\|)$.

# B IMPLEMENTATION

This section graphically describes the learning procedure in Algorithm 1. It also provides the training details not exposed in the main body of the paper.

## B.1 LEARNING SCHEME

We present in Figure 7 the updated graphics of our approach, highlighting the main components : the distributional part is ensured by a first deep neural network as is each of the mappings. As each of the networks should be learned, we included the part of the loss functions associated with each network (blue fonts correspond to minimization, whereas red fonts correspond to maximization, see Algorithm section).

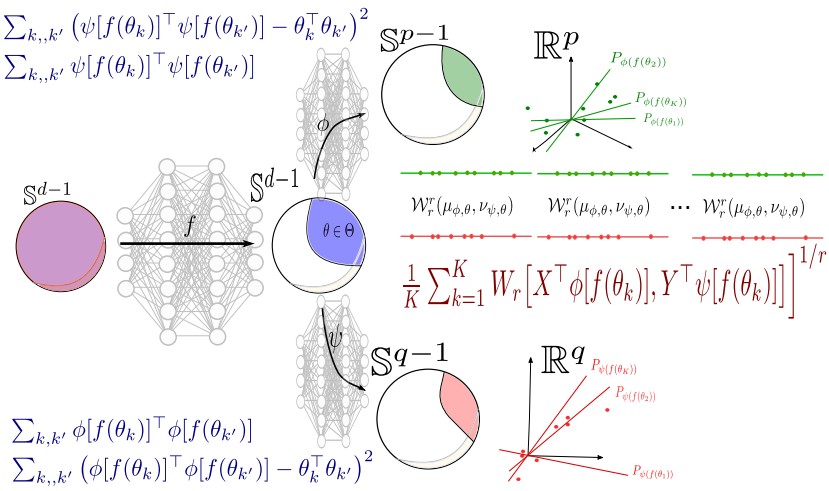

Figure 7: The implemented approach. Both the distributional and mappings parts are achieved by deep neural networks. A number $K$ of projections is used to compute 1D-Wasserstein distances.

## B.2 TRAINING DETAILS

Our experimental evaluations on shape datasets for scalability contrast GW, SGW and HWD. Regarding or classification under isometry transformations, we additionnaly consider RI-SGW. Used

hyper-parameters for those experiments are detailed below. Notice that SGW, RI-SGW and HWD rely on $K$, the number of projections sampled uniformly over the unit sphere. This $K$ may vary from a method to another.

1. SGW: $K$.
2. RI-SGW: $\lambda_{\text{RI-SGW}}$, the learning rate and $T$, the maximal number of iterations for solving 4 over the Stiefeld manifold.
3. HWD: beyond $K$ and the latent space dimension $d$, it requires the parametrization of $\phi$, $\psi$ and $f$ as deep neural networks and their optimizers. For solving the min-max problem by an alternating optimization scheme we use $N$ inner loops and $T$ number of epochs.

For SGW and RI-SGW we use the code made available by their authors and cite the related reference Vayer et al. (2018) as they require. We use POT toolbox Flamary et al. (2021) to compute GW distance.

**Scalability**   This experiment measures the average running time to compute OT-based distance between two pairs of shapes made of $n$ 3D-vertices. 100 pairs of shapes were considered and $n$ varies in $\{100, 250, 500, 1000, 1500, 2000\}$.

We choose $K_{\text{SGW}} = 1000$ (as a default value).

For HWD, the mapping function $f$ is designed as a deep network with 2 dense hidden layers of size 50. Regarding both $\phi$ and $\psi$, they have also the same architecture as $f$ (with adapted input and output layers) but the hidden layers are 10-dimensional. Adam optimizers with default parameters are selected to train them. Finally we consider $K_{\text{HWD}} = 10$, $d = 5$, $T = 50$, $N = 1$ as default values. Notice also that the regularization parameters $\lambda_C$ and $\lambda_a$ are set to 1.

The used ground cost distance for GW distance is the geodesic distance.

**Classification under transformations invariance**   For this experiment, we consider the same set of hyper-parameters as for **Scalability** evaluation on shape datasets. Besides, the supplementary competitor RI-SGW was trained by setting $K_{\text{RI-SGW}} = 1000 = K_{\text{SGW}} = 1000$, $\lambda_{\text{RI-SGW}} = 0.01$, $T_{\text{RI-SGW}} = 500$. Notice that due to the high-resolution of the meshes (more than 19K three-dimensional vertices), RI-SGW and GW were not able to produce the pairwise-distance matrix used in 1NN classification after several hours.

## C   ADDITIONAL EXPERIMENTAL RESULTS

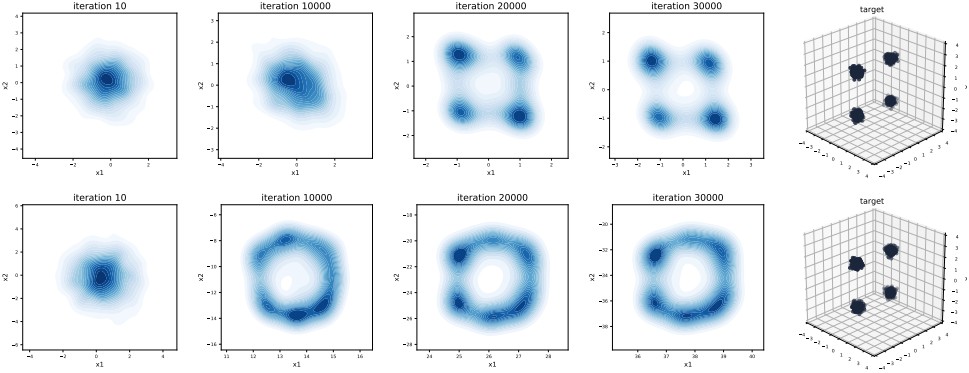

Figure 8: Comparing (top) HWD and (bottom) SGW on generating 2D distributions from 3D target.

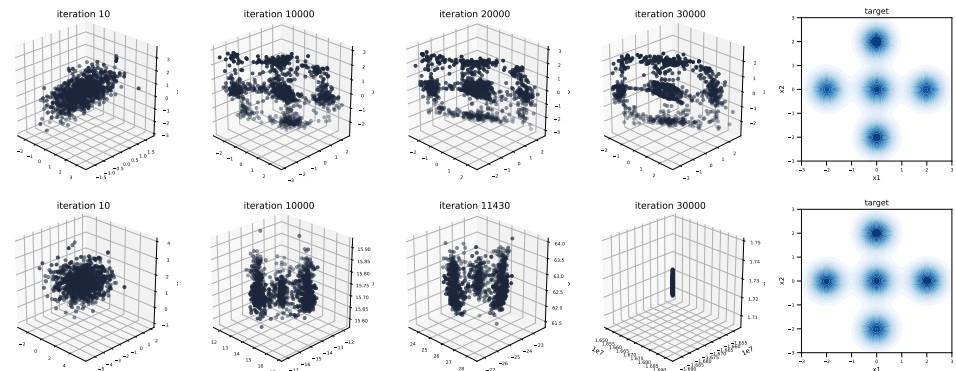

Figure 9: Comparing (top) HWD and (bottom) SGW on generating 3D distributions from 2D target.

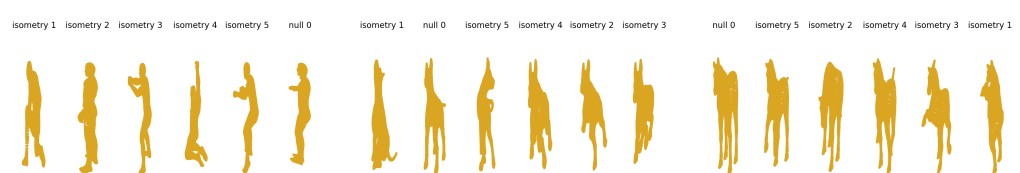

Figure 10: Instances of the shape dataset with null and isometry transformations. The classes are respectively `human`, `dog` and `horse`. For the experiments of Figure 6 we also consider the "topology", "scale", "shotnoise" transformations that respectively amount to deform, to upscale and to add noise to the shapes of each class.

