# OpenReview forum: "Heterogeneous Wasserstein Discrepancy for Incomparable Distributions"
_ICLR.cc/2022/Conference — ICLR 2022 Submitted_

### Official Review · Reviewer_rgFL · 2021-11-02

**Correctness:** 3
**Technical Novelty And Significance:** 1
**Empirical Novelty And Significance:** 2
**Recommendation:** 3
**Confidence:** 5

**Main Review:**

Here are my comments on the paper:

(1) The novelty of the paper is indeed limited. The HWD is a direct application of the idea of [1] Nguyen et al. [2020]. The formulation as well as the theoretical results are also direct from this work. Unless the authors can convince me in the rebuttal that they have something fundamentally new in their paper, the paper does not appear novel to me.

(2) Definition 1 is basically distributional sliced Wasserstein in [1] by projecting $\mu$ and $\nu$ into the same unite sphere. Furthermore, it seems to me that Proposition 1 also follows directly from Appendix B in [1]. The authors need to explain the novelty in their results.

(3) I have trouble understanding the implications of results in Proposition 2. The authors may need to explain further the meaning of each result in the rebuttal.

(4) The experiments are also mostly with toy examples, which limit the scope of the method. Do the authors also have the experimental results for real data?




Reference:

[1] Nguyen et al. [2020]. Distributional sliced Wasserstein distance and applications to deep generative model. ICLR, 2021.

**Summary Of The Paper:**

In the paper, the authors proposed heterogeneous Wasserstein discrepancy (HWD) to compare incomparable distributions. The main idea of HWD is via distributional slicing, which had been employed in the work of Nguyen et al. [2020].

**Summary Of The Review:**

In my opinion, the contribution of the paper is limited as it is a direct extension of [1]. Both the formulation of HWD and its theoretical results do not appear to be novel.

---

### Official Review · Reviewer_xwxk · 2021-11-02

**Correctness:** 2
**Technical Novelty And Significance:** 2
**Empirical Novelty And Significance:** Not applicable
**Recommendation:** 3
**Confidence:** 3

**Main Review:**

+ Strengths:

The authors propose a new variant of optimal transport for distributions supported in different spaces.

+ Weaknesses:

It seems that the contribution is quite incremental. It combines the approach in Alaya et al., 2020 to deal with distributions supported in different spaces via embedding for optimal transport, and the distributional sliced Wasserstein (Nguyen et al., 2020) to project into 1d-dimensional space to compute Wasserstein (instead of standard Wasserstein in some latent space in Alaya at al., 2020).

+ Detailed comments:

1. It seems that the main problem is that supports of distributions are living in different spaces. In case, one can learn the embedding as in Alaya et al., 2020 (e.g, about \phi, \theta embedding) -- which is also used as the main part of the proposed approach, the remaining problem is the optimal transport between distributions supported in some "latent" space. The latter part is the application of distributional sliced Wasserstein. Am I missing something?

2. As in the setting, distributions are supported in different spaces. So, even the dimension of those spaces are equal, they are still unaligned (e.g., different basic coordinates). So, for the 1d-projection, I wonder why the Wasserstein is favored than the Gromov-Wasserstein to align for those 1d-projected supports. (Imagine that the "arrow" object points to the positive infinity in 1d-space, but in other 1d-space, the "arrow" object points to the negative infinity in another 1d-space. Could you comment for such situations?)

3. For the Definition (5) for the proposed heterogeneous Wasserstein discrepancy (HWD), it seems that it is highly non-convex, and more "complicated" than the Gromov-Wasserstein (GW) in Equation (1)? E.g., HWD requires to optimize 2 embedding functions (\phi, \theta)? Could the authors comment why HWD is "intuitively" easier to solve than GW?

4. For complexity, why T=50 and N=1 are used? (especially N=1?) It seems that Algorithm 1 is alternating optimization?

5. For experiments about the translation (in Figure 2), is translation-invariant a good or bad property? (for distributions supported in different spaces)

6. For Figure 4, it seems "abrupt" for iteration 10K and iteration 20K, it is better in case the authors show more figures in that range (at least in the supplementary).

7. What are the stopping conditions for GW and RISGW in experiments?

8. In Figure 5, for time complexity, HWD is much faster than GW and is also faster than SGW when there are more than 1K meshes per each shape. Recall that the complexity of SGW is O(nlogn), it is unclear why HWD can be even faster? Am I missing something (HWD is highly non-convex, and requires to optimize two embedding functions together with the distribution for distributionally sliced-Wasserstein).



**Summary Of The Paper:**

The authors propose a variant of Wasserstein for distributions supported on different Euclidean spaces (e.g., different dimensions). The main ideas are to combine the embedding (for distributions supported in different spaces -- Alaya et al. 2020) and distributional sliced Wasserstein (Nguyen et al., 2020) to generate random slicing projections into 1d-space, and leverage the 1d-Wasserstein for the comparison. The authors apply the proposed method for generating modeling and in query framework.

**Summary Of The Review:**

It seems that the contribution is quite incremental (e.g., using embedding approach in Alaya et al. 2020 for distributions supported in different spaces to combine with distributional sliced-Wasserstein in Nguyen et al., 2020 to solve the Wasserstein in 1d-projected space).

---

### Official Review · Reviewer_MjvX · 2021-11-02

**Correctness:** 4
**Technical Novelty And Significance:** 3
**Empirical Novelty And Significance:** Not applicable
**Recommendation:** 6
**Confidence:** 3

**Main Review:**

The heterogeneous Wasserstein discrepancy as been proposed as a substitute to the classical Gromov-Wasserstein distance, that suffers from a high computational cost. Therefore, HWD is a computationally interesting tool to compare probability measures supported in spaces of different dimension. In particular, HWD is constructed in such a way that it is sensitive to translations whereas others methods based on slicing (such as Sliced Gromov-Wasserstein) are not.

Nevertheless, I find that the present paper could benefit from more explanations, insights and justifications.
To begin with, the Gromov-Wasserstein distance behaves fundamentally differently than the proposed HWD in the sense that it compares pairs of supports' points of distribution $\mu$ with pairs of supports' points of distribution $\nu$, whereas HWD compares directly the -projected- measures. Therefore, more comparison between GW and HWD, not only in terms of running time, but also theoretically and in terms of distance between shapes for example (which shapes are similar under GW, and which ones under HWD) is important in my opinion. Another point, in [Alaya et al., 2020], the proposed robust Wasserstein distance for measures in different spaces is defined in such a way to look for "the worst possible OT cost over low-distortion embeddings". A precise justification of the chosen formulation of the problem in this paper would be appreciated.

Additionally, it seems that the choice of $d$, the dimension of the latent space, is crucial, as large $d$ "focus on far-angles directions" and smaller $d$ "may lose the control on the distributional part". In particular, very little is said about how to properly choose $d$, and how restrictive actually are the sets $M_{C_\phi}$ and $M_{C_\psi}$; or equivalently, how restrictive is the angle preserving property on $\phi$ and $\psi$.

In addition, when defining the HWD, I am not sure that the compactness of the support of the measures is required, is it? Moreover, do $HWD_r(\mu,\nu)=0$ implies that $\mu=\nu$? And if not, in which case scenario this could occur, and what are the implications?

Computationally speaking, the convergence of the algorithm towards global optimizers is not guaranteed, and some discussion on that matter could be added.

Minor comments : At the end of page 13, in the first line of the last equation, the projection operator is missing for $\psi(\theta)$.

**Summary Of The Paper:**

This paper introduces the so-called Heterogeneous Wasserstein Discrepancy (HWD) between two probability measures supported on Euclidean spaces of different dimensions, a case scenario appearing in several applications. The HWD is built on ideas from the sliced Wasserstein distance and measure embedding : the two distributions are projected into a one-dimensional space, in which the classical Wasserstein distance (closed-form in 1D) is computed. Moreover, an algorithm to compute such a divergence is presented, and its usefulness is illustrated on synthetic examples, validated on real-world data and compared to standard others methods.

**Summary Of The Review:**

The HWD is a novel discrepancy for distributions on heterogeneous spaces, based on slicing and embeddings, and in this sense, is not very original. That being said, it is faster to compute than the traditional Gromov-Wasserstein and its approximations; and HWD provides convincing results as a discrepancy in generative models. Nevertheless, I think that the present paper lacks of justifications on (i) how this discrepancy behaves, (ii) the formulation of the problem, (iii) the choice of the hyperparameter $d$, (iv) the role played by the sets of constraint $M_{C_\phi}$ and $M_{C_\psi}$.

---

### Decision · Program_Chairs · 2022-01-20

**Decision:**

Reject

**Comment:**

The aim of this paper is to propose a novel "GW"-like discrepancy function between probability measures living in different spaces (here restricted to be Euclidean, with a squared euclidean distance as the base metric). While interesting (notably the idea of learning distinct maps mapping a random direction in a latent space onto two spaces) there are a few issues with presentation, incremental nature of work and importantly a few shortcomings in the empirical evaluation as detailed by reviewers. Hopefully these can be used to improve the draft for a future version.